# Ecological stoichiometric characteristics of soil-moss C, N, and P in restoration stages of karst rocky desertification

Wenping Meng[1,2,3], Quanhou Dai[1,4]*, Qingqing Ren[5], Na Tu[1,4], Tingjiao Leng[1,4]

**1** College of Forestry, Guizhou Universrty, Guiyang, China, **2** Guizhou Botanical Garden, Guiyang, China, **3** Puding Karst Ecosystem Research Station, Institute of Geochemistry, Chinese Academy of Sciences, Guiyang, China, **4** Institute for Forest Resources & Environment of Guizhou, Guiyang, China, **5** Affiliated Hospital of Zunyi Medical University, Zunyi, China

* qhdairiver@163.com

**Data Availability Statement:** All relevant data are within the manuscript and its Supporting Information files.

**Funding:** This work was supported by National Natural Science Foundation of China (41671275),

## Abstract

Rocky desertification is the most serious ecological disaster in karst areas. Comprehensive control of rocky desertification plays an important role in promoting the economic development of karst areas. Studying the stoichiometric characteristics of mosses and soil can provide a powerful reference for the ecological restoration and evaluation of ecosystems experiencing rocky desertification. Soil and mosses were collected from sites representing different stages of ecological restoration (bare rock, grassland, shrubland, and secondary forest), and the contents of carbon (C), nitrogen (N), and phosphorus (P) were detected for ecological stoichiometric analysis. The results indicate that in different restoration stages following karst rocky desertification, the contents of soil organic carbon (SOC), total nitrogen (TN), and total phosphorus (TP) and the stoichiometric ratios in the shrub habitat are higher than those in the bare rock, grassland, and secondary forest habitats. However, the TP and available P contents were low at all stages (0.06 g/kg and 0.62 mg/kg, respectively). The N and P contents and stoichiometric ratios in the mosses showed no significant differences among the succession stages. The C contents in the mosses had a significant positive correlation with SOC and TN and TP content, and the P content had a significant positive correlation with the soil available P. However, there was a significant negative correlation between the C: N and C:P ratios of the bryophytes and soil C: N. In summary, during the process of natural restoration of karst rocky desertification areas, SOC and soil TN contents accumulate with each succession stage. Soil nutrients are higher in shrub habitats than in other succession stages. Mosses have a strong effect on improving soil nutrients in rocky desertification areas.

## Introduction

Karst rocky desertification is a process of land degradation caused by the combined effects of natural factors and human activities in the fragile karst background of the subtropics [1, 2]. It manifests as the destruction of vegetation, soil erosion, decline in land productivity, and large

Science and Technology Fund of Guizhou ([2020] 1Y074), National Nature Science Foundation of China and the Karst Science Research Center of Guizhou Province (U1812401), Youth Key Fund of Sciences Academy of Guizhou([2020]03), Forestry Research of Guizhou Province ([2019]06). The funding supporter provided the cost of purchasing experimental equipment, materials, literature, and investigation of this project.

**Competing interests:** The authors have declared that no competing interests exist.

**Abbreviations:** SOC, Soil organic carbon; TC, Total carbon; TN, Total nitrogen; TP, Total phosphorus; P, Phosphorus; N, Nitrogen; C, Carbon.

areas of bare rock, similar to desertification landscape [3, 4]. Rocky desertification has become one of the obstacles to sustainable ecological development in Southwest China [5–7]. In recent years, local conflicts between people and land have been alleviated, and the economy has developed through the planting of economic tree species [8]. However, with tree growth, nutrient requirements have increased annually, leading to diminished soil nutrients, which in turn deteriorate the soil environment in rocky desertification areas.

Ecological stoichiometry reflects the nutritional structure and function of an ecosystem by examining the balance between energy and chemical elements (essential elements such as C, N, and P) in the ecosystem [9–11]. In soil stoichiometry, C: N, C:P and N:P ratios are key indicators that reflect the composition of soil organic matter and the availability of soil nutrients [12, 13]. However, the nutrient cycle and ecological stoichiometry of the restoration process in karst rocky desertification are not well understood.

Mosses are one of the most widely distributed plants in the world [14]. Their special leaf surface structure and cell characteristics allow them to withstand high temperatures [15, 16] and drought, provide strong water storage capacity and moisture retention ability and stabilize soils [17, 18]. Mosses play an important role in preventing and controlling soil erosion on rock surfaces [19]. The H2CO3 formed by moss respiration and secretions can dissolve rocks and form primitive soil [20–22]. Additionally, organic matter secreted by mosses complexes with mineral ions and forms insoluble matter [23, 24]. Insoluble matter adheres to moss residue, which not only increases soil deposition but also promotes organic matter accumulation and increases soil nutrient contents [23, 25, 26]. Compared to bare soil, moss biocrusts were found to have a positive effect on all soil nutrients and to buffer the negative effects of karst rocky desertification, significantly increasing soil microbial richness [27]. Mosses are more sensitive to environmental changes than other plants and are often used for environmental monitoring [28].

Therefore, studying the stoichiometric characteristics of mosses and soil can reveal the nutrient cycle of topsoil during the natural restoration of areas that have undergone karst rocky desertification. This research provides new ideas and methods for controlling karst rocky desertification. Bare rocks, grasslands, shrubs, and secondary forests in different stages of natural restoration following karst rocky desertification were selected as the study area. Mosses and soil were collected from the study sites to detect C, N, and P contents, and the ecological stoichiometric characteristics were analysed.

## Material and methods

### Site description

The full name of the work site is the Puding Karst Ecosystem Research Station, Institute of Geochemistry, Chinese Academy of Sciences. The workstation is open to researchers, and no special permit is required to work in the station. The geographical location is 26˚22'07.06" north and 105˚45'06.65" east, and the study site has an altitude of 1175 m (Fig 1). It has a humid northern subtropical monsoon climate, with an average annual temperature of 15.1˚C, and the average annual precipitation is 1396.9 mm. Karst landforms occur widely in the area. The bedrock is mainly dolomite and limestone, the area of rocky desertification exceeds 80%, the soil is mainly lime soil and yellow soil, and the vegetation coverage is approximately 10% to 20%.

### Experimental design and field sampling

Bare land, grasslands, shrubs, and secondary forests in restoration areas of karst rocky desertification were chosen as plots (Table 1. Each plot consisted of a circle 400 m² in size. Mosses

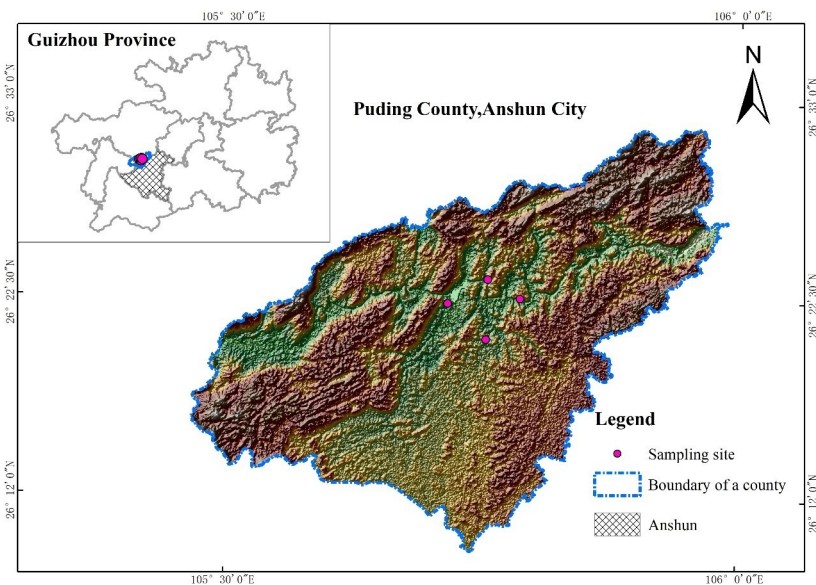

**Fig 1. A map of the study area and sampling plot.**

were randomly distributed in each plot (Fig 2). According to the moss collection method, each plot was divided into 125 small plots (circles 400 cm² in size). All the mosses and 1 cm of soil under the mosses in the small plots were collected. The collection number, time, habitat information and characteristics of the mosses were recorded. A total of 500 moss specimens and 500 soil specimens were collected.

## Material and calculational formula

After identifying the mosses species, the relative coverage and relative frequency of rocky mosses were used to calculate the ecological importance of the species. The dominant species

**Table 1. Habitat characteristics of the plot.**

| different restoration stages | latitude and longitude | slope | aspect | rock exposure rate | vascular plant species in the plot |
|---|---|---|---|---|---|
| bare rock | E 105°45′12″, N 26°22′03″ | 20° | E | 90% | *Celastrus gemmatusPaederia foetidaMelia azedarachFallopia multifloraPhytolacca americanaToxicodendron vernicifluumSetaria viridisCeltis sinensisMahonia fortunei Solanum americanumCyclosorus parasiticusAsplenium trichomanesDrynaria roosii* |
| grassland | E 105°21′50″, N 26°22′16″ | 30° | S | 40% | *Toona sinensisAlangium chinenseFallopia multifloraRubus corchorifoliusMiscanthus sinensis Justicia procumbensAgeratum conyzoidesPteris multifidaPilea cavalerieiTalinum paniculatumMosla scabraMentha canadensisDigitaria sanguinalis* |
| shrub | E 105°45′04″, N 26°22′03″ | 60° | SE | 50% | *Ilex macrocarpaAlangium chinenseAgeratina adenophoraCuscuta chinensisDebregeasia orientalisFissistigma chloroneurumAkebia trifoliata subsp. australisRubus ellipticusMelia azedarachSmilax chinaCaesalpinia cristaRosa cymosaDalbergia assamicaLitsea coreana var. sinensisRubus rosifoliusRubus tephrodesNeolepisorus fortuneiPyrrosia calvataPilea cavaleriei Drynaria roosiiCladrastis platycarpaCeltis sinensisLindera communisNandina domestica Rhus chinensisRhynchosia volubilis* |
| secondary forest | E 105°45′04″, N 26°22′03″ | 40° | SW | , 60% | *Castanea mollissimaCladrastis platycarpaNandina domesticaBroussonetia papyriferaRhus chinensisCeltis sinensisCaesalpinia cristaMallotus repandus var. chrysocarpusBauhinia championiiFallopia multifloraZanthoxylum dissitumPyracantha fortuneanaDebregeasia orientalisDigitaria sanguinalisLitsea coreana var. sinensisSporobolus fertilisTriadica sebifera Rosa cymosaAkebia trifoliataMiscanthus sinensisAgeratina adenophoraToona sinensis Lespedeza cuneataParthenocissus tricuspidata* |

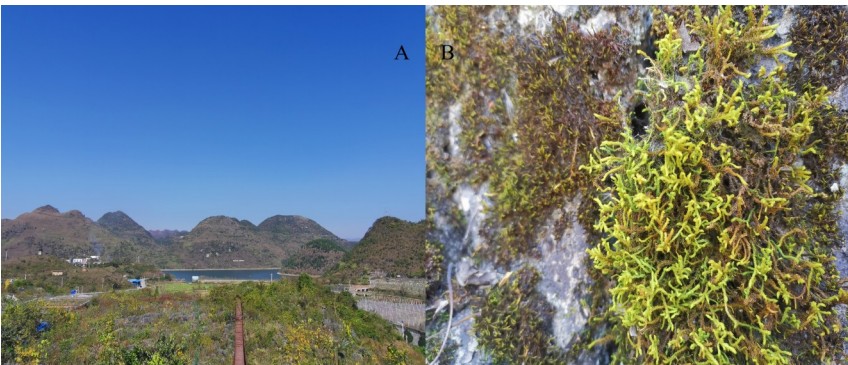

**Fig 2. The map of the ecosystem of Puding karst rocky desertification ecosystem observation and research station of the chinese academy of sciences.** A show the habitat of karst rocky desertification. B shows mosses in the karst rocky desertification habitat.

of mosses in the rocky desertification areas were selected as study species, and their C, N, and P contents were analysed.

The relevant formulae are as follows:

1. Frequency = (the number of plots containing a certain mosses/the total number of plots) ×100%

2. Relative frequency = (frequency of a certain mosses/frequency of all mosses present) × 100%

3. Coverage = (coverage of a certain mosses/square area) × 100%

4. Relative coverage = (coverage of a certain bryophyte/coverage of all species of mosses) × 100%

5. Ecological importance value = (phase frequency + relative coverage)/2

## Determination of C, N and P elements in soil and mosses

The soil samples collected in this study were analysed to determine the SOC, TN, TP and available phosphorus (Olsen-P). SOC was measured using the external heating method. To determine the SOC content, potassium dichromate and sulfuric acid solution were added to air-dried soil samples. The solution was heated in a pan with paraffin oil, boiled for 5 minutes, and titrated with ferrous sulfate solution after cooling, after which the amount of consumed ferrous sulfate was determined. From the amount of consumed ferrous sulfate, the SOC content was calculated. The contents of TN, TP and available phosphorus (Olsen-P) were measured by using previously published methods described by Bao [29].

The C in the mosses was directly determined by an elemental analyser (Elementar Vario TOC, Germany). Both N and P were digested by concentrated sulfuric acid and hydrogen peroxide. The moss samples were digested in concentrated sulfuric acid until the sulfuric acid decomposed and white smoke was emitted. When the solution was brown and black, it was cooled, hydrogen peroxide was added, and the mixture was boiled for 20 minutes. This process was repeated several times until the digestion liquid was colourless and clear. The liquid was then heated for another 10 minutes to remove excess hydrogen peroxide and cooled to a constant volume. A flow injection instrument (AA3) was used for direct measurement.

## Statistical analysis

Software such as Excel 2013 and SPSS 22.0 were used for statistical analysis of the data, and Origin 8.0 software was used for graphing. The C:N, C:P, and N:P stoichiometric ratios of soil and mosses are expressed by mass ratios. One-way analysis of variance (one-way ANOVA) and Duncan's method were used to examine the C, N, and P contents and C:N, C:P, and N:P stoichiometric ratios of soil and mosses at different stages of recovery from karst rocky desertification ($\alpha = 0.05$). Pearson correlation analysis was used to determine the relationships between soil and bryophyte C, N, and P contents. The data are expressed as the average ± standard deviation.

# Results

## Soil C, N, and P contents and their stoichiometric ratios in different vegetation restoration stages

The average contents of SOC, TN and TP were 133.35 g/kg, 9.99 g/kg, and 0.06 g/kg, respectively (Table 2). Soil nutrients gradually accumulate with the restoration of karst rocky desertification. However, the contents of SOC, TN, and TP in the shrubland were the highest, and they were significantly different from those in the bare rock, grassland and secondary forest sites (P<0.05). The average content of Olsen-P was 0.62 g/kg, and its change trend was different from that of TP. The highest content of Olsen-P was found in the bare rock habitat (0.732 ±0.25 g/kg), and this value was significantly different from that in the other three habitats (P<0.05).

The average C:N ratio of soil in the rocky desertification areas was 14.83. With ecosystem succession, the C:N ratio gradually decreased. The C:N ratio in the bare rock site was as high as 19.01±7.61, which was significantly different from that observed for the other three restoration stages (P<0.05). The trends of the changes in C:P and N:P with habitat were basically the same, and the ratios of the bare rock and grassland habitats were significantly different from those of the other habitats (P<0.05).

## The contents of C, N, and P in mosses at different vegetation restoration stages and their stoichiometric ratios

The average C, N and P contents of mosses were 3.15 g/kg, 13.97 g/kg and 3.20 g/kg, respectively, in the karst rocky desertification areas (Table 3). The moss C content changed with restoration from karst rocky desertification and was significantly different between different recovery periods (P<0.05). Among the bare rock, grassland, shrub and secondary forest sites, the N and P contents of bryophytes were the highest in the secondary forest, with values of

**Table 2. The contents and stoichiometric ratios of soil C, N, and P in different stages of restoration from karst rocky.**

| different restoration stages | SOC (g/kg) | TN (g/kg) | TP (g/kg) | Olsen-P (mg/kg) | C: N | C:P | N:P |
|---|---|---|---|---|---|---|---|
| bare rock | 92.91±8.51c | 5.39±1.47c | 0.06±0.01b | 0.732±0.25a | 19.01±7.61a | 1589.38±268.98b | 95.85±38.52c |
| grassland | 65.89±20.39d | 4.39±1.45c | 0.05±0.02c | 0.57±0.06b | 15.26±2.88b | 1521.54±648.65b | 98.89±32.73c |
| shrub | 224.89±9.74a | 18.16±1.03a | 0.07±0.01a | 0.61±0.08ab | 12.42±0.85b | 3160.25±572.19a | 253.97±40.76a |
| secondary forest | 149.72±60.37b | 12.00±4.42b | 0.06±0.01bc | 0.57±0.11b | 12.62±2.70b | 2652.69±1066.88a | 207.73±69.01b |
| average value | 133.35 | 9.99 | 0.06 | 0.62 | 14.83 | 2230.97 | 164.11 |

Different lowercase letters in the same column indicate significant differences between different habitats (P<0.05), and P in the stoichiometric ratio represents TP.

**Table 3. The contents and stoichiometric ratios of C, N, and P in mosses at different stages of recovery from karst rocky desertification.**

| different restoration stages | C (g/kg) | N (g/kg) | P (g/kg) | C:N | C:P | N:P |
|---|---|---|---|---|---|---|
| bare rock | 2.99±0.20a | 14.42±3.83a | 3.12±0.48ab | 0.22±0.05b | 1.21±0.54b | 5.28±1.44b |
| grassland | 2.57±0.29b | 9.67±5.42b | 3.18±0.47ab | 0.54±0.57a | 1.02±0.50b | 4.15±2.49b |
| shrub | 3.84±0.36c | 15.65±5.77a | 2.47±0.39b | 0.28±0.10b | 2.42±1.97a | 9.16±6.47a |
| secondary forest | 3.22±0.15d | 16.14±3.59a | 4.04±0.62a | 0.21±0.05b | 1.00±0.45b | 5.04±2.97b |
| average value | 3.15 | 13.97 | 3.20 | 0.31 | 1.42 | 5.91 |

Different lowercase letters in the same column indicate significant differences between different habitats (P<0.05).

16.14±3.59 g/kg and 4.04±0.62 g/kg, respectively; these values were significantly different from those of the other sites (P<0.05).

The average C:N stoichiometric ratio of the mosses was 0.31. The C:N ratio of the bare rock, shrubs and secondary forests tended to be stable, while that of the grassland was not. The average C:P and N:P stoichiometric ratios of the mosses were 1.42 and 5.91, respectively. Among all the habitats, the C:P and N:P stoichiometric ratios were the highest in the shrubs, and these values and were significantly different from those at other successional stages (P< 0.05). The C:P and N:P stoichiometric ratios in the bare rock, grassland and secondary forest habitats tended to be stable.

## The relationships between soil and bryophyte C, N, and P contents and stoichiometric ratios

Correlation analysis revealed that SOC and soil TN were significantly positively correlated with TP and the C:P and N:P ratios. The correlations between soil TP, Olsen-P and the N:P ratio were not significant. TP was negatively correlated with the C:P and C:N ratios. Olsen-P was not correlated with SOC, TN, or the C:P or N:P ratios. The C:N ratio had a significant positive correlation with Olsen-P and significant negative correlations with SOC and TN. There was a significant negative correlation between the C:N and N:P ratios, no correlation between the C:N and C:P ratios, and a significant positive correlation between the N:P and C:P ratios (Table 4).

There was a significant positive correlation between C and N in mosses, but neither was related to P. C and N had significant negative correlations with the C:N ratio. C had significant positive correlations with the C:P and N:P ratios. P had a significant negative correlation with the C:P and N:P ratios but was not correlated with the C:N ratio. N had a significant positive

**Table 4. Pearson correlation analysis between C, N, and P contents and stoichiometric ratios of karst rocky desert soil.**

| Soil | SOC | TN | TP | Olsen-P | C: N | C:P | N:P |
|---|---|---|---|---|---|---|---|
| SOC | 1 | | | | | | |
| TN | .961** | 1 | | | | | |
| TP | .492** | .502** | 1 | | | | |
| Olsen-P | 0.075 | -0.096 | 0.209 | 1 | | | |
| C:N | -.289* | -.496** | -0.048 | .658** | 1 | | |
| C:P | .843** | .778** | -0.013 | -0.016 | -0.242 | 1 | |
| N:P | .878** | .899** | 0.115 | -0.143 | -.501** | .937** | 1 |

*. Correlation is significant at the 0.05 level (2-tailed)

**. Correlation is significant at the 0.01 level (2-tailed), The P in the stoichiometric ratio is calculated by TP content.

**Table 5. Pearson correlation analysis of the C, N, and P contents and stoichiometric ratios of mosses.**

| Mosses | C | N | P | C: N | C:P | N:P |
|--------|-----|-----|-----|------|-----|-----|
| C | 1 | | | | | |
| N | .587** | 1 | | | | |
| P | -0.263 | 0.075 | 1 | | | |
| C:N | -.417** | -.739** | 0.264 | 1 | | |
| C:P | .508** | 0.023 | -.667** | -0.144 | 1 | |
| N:P | .586** | .310* | -.685** | -.330* | .935** | 1 |

*. Correlation is significant at the 0.05 level 2-tailed

**. Correlation is significant at the 0.01 level (2-tailed), The P in the stoichiometric ratio is calculated by TP content.

correlation with the N:P ratio but was not correlated with the C:P ratio. The stoichiometric ratios of the mosses showed a significant positive correlation between the C:P and N:P ratios, a significant negative correlation between the C:N and N:P ratios, and no correlation between the C:P and C:N ratios (Table 5).

The content of C in mosses was significantly positively correlated with SOC (r = 0.766, P<0.01), soil TN (r = 0.795, P<0.01) and soil TP (r = 0.485, P<0.01) but not with soil Olsen-P. The content of N in mosses was significantly positively correlated with SOC (r = 0.329, P<0.05) and soil Olsen-P (r = 0.338, P<0.05), and there were no significant correlations between the N contents in the mosses and soil N and P contents. The P content in the mosses was significantly positively correlated with soil Olsen-P (r = 0.433, P<0.01) but not significantly correlated with soil C, N, or P contents (Fig 3).

There were significant negative correlations between the moss C:N and C:P ratios and the soil C:N ratio, a negative correlation between the bryophyte N:P ratio and the soil C:N ratio, and a negative correlation between the C:P ratio and N contents of the mosses and soil. There was a positive correlation between the P contents of the mosses and soil, but the correlation between the two was not significant. In addition, other stoichiometric correlations between the mosses and soil were not significant (Fig 4).

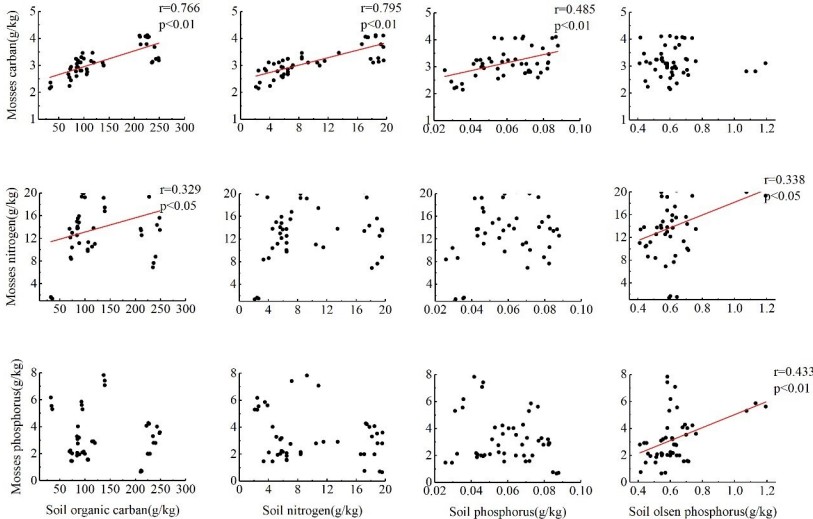

**Fig 3. Correlation analysis between mosses and soil C, N, and P contents in karst rocky desertification areas.**

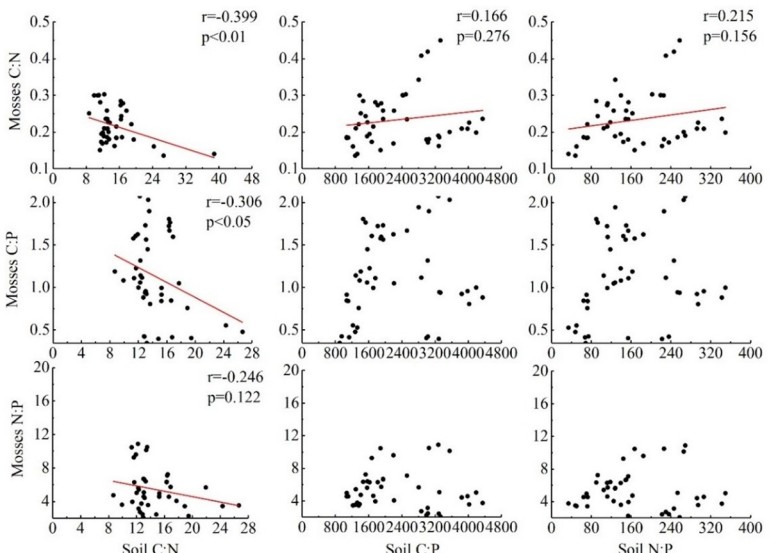

**Fig 4. Correlation analysis between C, N, and P contents of mosses and soil and stoichiometric ratios in karst rocky desertification areas.**

## Discussion

### Soil C, N, and P contents and stoichiometric characteristics of different restoration stages of karst rocky desertification areas

The contents of SOC, soil TN and soil TP in the karst rocky desertification area did not strictly follow the pattern of succession but showed a stepped increasing trend. he contents of SOC, soil TN and soil TP in the shrub habitat were the highest, followed by the contents in the secondary forest. These results are consistent with the research of Li [30]. The consistency between results may be because the pioneer shrubs present during rocky desertification restoration were mainly Rosa cymosa, Rubus corchorifolius, Akebia trifoliata, Cladrastis platycarpa, etc. These plants produce mainly papery leaves, which decompose easily. The pioneer tree species in the secondary forest were mainly Litsea coreana and Castanea mollissima; these species generally produce leathery leaves that take a long time to decompose.

The nutrient requirements of arbour tree species are greater than those of other vegetation types [31]. With the restoration of vegetation, the nutrient storage rate of the community decreases, the nutrient cycle accelerates, and the nutrient turnover time is long in the middle-high bud subclimax community stage [32], resulting in the soil nutrient content at the shrub site being significantly higher than that at the secondary forest site.

Notably, this study showed that the average SOC and soil TN contents of mosses in the shrub habitat were 133.35 g/kg and 9.99 g/kg, respectively; these values were significantly higher than the SOC and soil TN in the vegetated areas of the Maolan karst forest (54.72 g/kg and 4.67 g/kg) [33] and karst rocky desertification-affected secondary forests (80.40 g/kg and 2.80 g/kg) [32]. Mosses have a slower decomposition rate than vascular plants, resulting in high organic matter content in moss substrates [34]. Moreover, mosses can form symbiotic relationships with blue algae [35]. Moss-cyanobacteria symbiosis can lead to more efficient N fixation and transport on the soil surface of forests [36], resulting in a high N content in the moss substrate. N fixed by mosses is an important pathway of N sources and sinks for forest ecosystems [37], which has ecological significance that cannot be ignored for ecosystems and even the global nitrogen input and cycle [38].

The soil C:N ratio reflects the soil fertility level and the decomposition rate of soil organic matter [39]. A lower C:N ratio represents high fertility and faster C and N mineralization rates [40]. This study found that the secondary forests and shrubs had lower soil C:N ratios than the other habitat types. The soil C:N ratio reflects the level of soil fertility and the decomposition rate of soil organic matter [41]. Generally, a lower soil C:N ratio represents high fertility and faster C and N mineralization [42]. This shows that under the natural restoration of rocky desertification, soil fertility gradually increases with the succession of the ecosystem. The availability of P is determined by the decomposition rate of soil organic matter, and a lower C:P ratio is an indicator of higher P availability [43]. During the process of restoration from rocky desertification, the C:P ratio tends to increase with succession. The availability of P gradually decreases with succession in the ecosystem. This may be due to the increase in biodiversity as succession advances and the composition of soil nutrients becoming more complicated, both of which limit the availability of P. Therefore, ways to improve soil fertility and promote the sustainable development of soil productivity should be considered in the comprehensive management of rocky desertification via ecological restoration.

## C, N, and P contents and stoichiometric characteristics of mosses in karst rocky desertification areas

The differences in the stoichiometric ratios of the key nutrient elements, such as C and N and P, in biomass can regulate and affect the process of carbon consumption or fixation in an ecosystem [44]. The change trend of the C contents in mosses is consistent with the SOC contents in karst rocky desertification areas and significant differences between different recovery stages. However, the P contents in the mosses did not differ greatly between the different recovery stages.

It is worth noting that the average P and N contents in the mosses (3.20 g/kg and 13.97 g/kg, respectively) were higher than the P and N contents of other plants (0.30 g/kg and 6.96 g/kg, respectively) in karst areas [45]. Studies have found that the absorption of P by plants is limited by water, and the use of water is limited by P [46]. Leaf P content is significantly positively correlated with annual average precipitation and precipitation during the growing season [47]. There is no waxy cuticle on the leaves and stems of mosses, and the moss surface can absorb water, nutrients and other substances in the atmosphere [48]. The dry and wet deposition of N and P elements in the atmosphere is also the reason why the N and P contents of mosses are higher than those of other plants. There are differences in the characteristics of the leaf P composition of different plants and functional groups [49]. The morphological structure of mosses is different from that of other plants, and mosses may have special regulatory physiological processes for P.

The growth rate hypothesis posits that changes in the growth rate cause changes in the stoichiometric C, N, and P ratios of organisms [50]. Plants with high growth rates usually have lower C:N, C:P, and N:P ratios, so P is allocated to ribosomal RNA to meet the requirements for rapid synthesis of protein by ribosomes to support rapid plant growth [51, 52]. Mosses are different from other plant groups in photosynthetic C fixation and the demand and utilization efficiency of nutrient elements [53]. It is necessary to study the ecological stoichiometric characteristics of moss C, N, P and other elements and establish a theoretical system applicable to moss ecological stoichiometry.

## Contents and chemical characteristics of soil-bryophyte C, N, and P in different restoration stages after karst rocky desertification

Previous studies have revealed many geometric relationships between the contents of C, N, and P in plant leaves, such as the constant or allometric growth between the N and C contents

of leaves and the exponential growth pattern of leaf P content at 3/4 of the growth of the C content [54, 55]. This study found that the contents of C and N in mosses were extremely significantly positively correlated ($P<0.01$), indicating a certain geometric relationship between the demands for C and N in mosses during the growth process. The contents of N and P in mosses were significantly higher than those in soil. This result was consistent with the high N and P contents in the leaves and litter of five subalpine forest types in central Yunnan [56]. Moreover, the stoichiometric ratios of soil C, N, and P were also significantly higher than those of mosses. This phenomenon is different from the ecological stoichiometric ratios between trees, shrubs and soil [57, 58]. It is worth noting that the N content of mosses is not related to the soil TN content but is significantly positively correlated with the SOC and Olsen-P. This shows that the N absorption of mosses is not directly limited by the soil N content, which may be caused by the special ecological habits and growth characteristics of mosses.

Many studies have shown that there are no correlations or weak correlations between the C, N, and P ratios of most plant leaves and the C, N, and P ratios of soil [59]. It is generally thought that the C, N, and P ratios of plant leaves are determined by the characteristics of species, including their environmental adaptability, rather than by soil nutrient limitations [60]. In this study, the C:N, C:P, and N:P ratios in mosses were negatively correlated with the soil C: N ratio. Among them, the C:N and C:P ratios of mosses were significantly negatively correlated with the soil C:N ratio, while the correlations between the other stoichiometric ratios were either not significant or weak. This result shows that the absorption of N and P by mosses is negatively correlated with the N content in the soil. When the N content in the soil is low, the N and P contents in mosses are higher.

## Conclusion

During the natural restoration of karst rocky desertification, SOC and TN contents accumulate with succession. Soil nutrients are higher in areas dominated by shrubs than in other succession stages. The C:N, C:P and N:P stoichiometric ratios increase with the succession of the ecosystem and tend to be stable at the sub-climax community stage. The contents of C, N and P in mosses and their substrates are higher than those in vascular plants and their substrates. The application of mosses could be used as a supplementary method to control karst rocky desertification and promote the sustainable development of the local economy due to their positive effects on improving soil nutrients.

## Supporting information

**S1 Text. Title is Stoichiometric data in moss and soil.**
(XLSX)

## Author Contributions

**Conceptualization:** Wenping Meng, Quanhou Dai.

**Data curation:** Qingqing Ren.

**Investigation:** Na Tu, Tingjiao Leng.

**Methodology:** Na Tu.

**Resources:** Qingqing Ren.

**Software:** Wenping Meng.

**Writing – original draft:** Wenping Meng.

**Writing – review & editing:** Wenping Meng, Quanhou Dai.

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
