## [Decision Letter · Decision Letter 0]

17 Mar 2021

PONE-D-21-06846

Ecological stoichiometric characteristics of soil-moss C, N, and P in restoration stages of karst rocky desertification

PLOS ONE

Dear Prof. Dai,

Thank you for submitting your manuscript to PLOS ONE. After careful consideration, we feel that it has merit but does not fully meet PLOS ONE’s publication criteria as it currently stands. Therefore, we invite you to submit a revised version of the manuscript that addresses the points raised during the review process.

Please submit your revised manuscript in four weeks. If you will need more time than this to complete your revisions, please reply to this message or contact the journal office at plosone@plos.org. Please include the following items when submitting your revised manuscript:

We look forward to receiving your revised manuscript.

Kind regards,

Fuzhong Wu

Academic Editor

PLOS ONE

Journal Requirements:

'This work was supported by National Natural Science Foundation of China (41671275), National Nature Science Foundation of China and the Karst Science Research Center of Guizhou Province (U1812401), Science and

Technology Fund of Guizhou ([2020]1Y074), Youth Key Fund of Sciences Academy of

Guizhou([2020]03), Forestry Research of Guizhou Province ([2019]06).'

'NO - Include this sentence at the end of your statement: The funders had no role in study design, data collection and analysis, decision to publish, or preparation of the manuscript.'

6. We note that Figure 1 in your submission contain [map/satellite] images which may be copyrighted. All PLOS content is published under the Creative Commons Attribution License (CC BY 4.0), which means that the manuscript, images, and Supporting Information files will be freely available online, and any third party is permitted to access, download, copy, distribute, and use these materials in any way, even commercially, with proper attribution. For these reasons, we cannot publish previously copyrighted maps or satellite images created using proprietary data, such as Google software (Google Maps, Street View, and Earth). For more information, see our copyright guidelines: http://journals.plos.org/plosone/s/licenses-and-copyright.

Additional Editor Comments (if provided):

Reviewers' comments:

Reviewer's Responses to Questions

**Comments to the Author**

1. Is the manuscript technically sound, and do the data support the conclusions?

Reviewer #1: Partly

Reviewer #2: Yes

2. Has the statistical analysis been performed appropriately and rigorously? 

Reviewer #1: Yes

Reviewer #2: Yes

3. Have the authors made all data underlying the findings in their manuscript fully available?

Reviewer #1: No

Reviewer #2: Yes

4. Is the manuscript presented in an intelligible fashion and written in standard English?

Reviewer #1: Yes

Reviewer #2: Yes

5. Review Comments to the Author

Reviewer #1: This manuscript provides a study on the ecological stoichiometric characteristics of bryophytes and soil under different stages of ecological restoration (bare rock, grassland, shrubland, and secondary forest) in karst areas. The topic fits well with the aim and scope of the journal. The topic is also interesting as it can show the nutrient limitation and utilization efficiency of bryophytes in different stages of ecological restoration. However, the present manuscript has some problems, one being that the Engling needs a revision, ideally by a native English-speaking person. Another one was also the logic development in the introduction and discussion sections were not clear among different paragraphs, and the sentences were some awkward and repetitive. Additionally, there is an unclear situation as regards the section of Material and methods. Detailed comments follow below.

Introduction:

Although the author introduced in detail why ecological restoration should be carried out in karst areas, the current manuscript did not clearly explain why ecological stoichiometric characteristics of bryophytes and soil should be studied in different ecological restoration stages. Why this study is important now and novel compared to the previous research? Similarly, what are the scientific questions of this study? In short, I am not clear about the innovative and scientific purpose of this study.

Material and methods:

The authors spent a lot of effort to describe the materials and methods of this study, but unfortunately, the key contents were not shown in the MS. As a reader, I would like to know more about the plant communities in different stages of ecological restoration, such as the geographic information, dominant species, etc. Additionally, it is surprising that the author did not describe the depth of the soil samples collected. These make it difficult for me to make a proper judgment on the results of the study.

Results:

There are two serious problems. One is that the language is lengthy, and the description of results still unclear. Another one is interesting that the N and P concentrations were higher C concentrations in the bryophytes under all stages of ecological restoration. Although bryophytes have unique biological characteristics compared with other plants, the current results are not convincing. Of course, if it was not for the miscalculation, I would like the authors will have an appropriate explanation for this phenomenon in the discussion section.

Discussion and Conclusion:

The discussion is mostly descriptive or compares the results with others, without much explanation on how the data they use reflects the different ecophysiological process taken place in different stages of ecological restoration.Thus, it is better to give more explanations to the difference of bryophytes (or soil) ecological stoichiometry in different stages of ecological restoration rather than to extensive discussion. So for example, why the soil C, N, and P concentrations were higher in shrub? What are the implications of these results for ecological restoration? Additionally, be concise on your findings and delete unnecessary details and speculative in the Conclusion section.

Specific comments

Lines 54-66, Please use the corresponding abbreviation of carbon, nitrogen, and phosphorus in the Abstract.

Line 65, what is meant with “other ecological restoration”? how did you come to this conclusion or opinion?

Lines 114-117, it is not clear how to promote the accumulation soil and enhancement of nutrient due to study the ecological stoichiometric.

Lines 120-122, please check whether this sentence is correct?

Lines 138-143, it is not clear what the authors want to say here.

Lines 143-145, This information is very important to the results, please add relevant information.

Lines165-168, please check whether the test method is correct.

Lines 203-204, The sentence is incomplete.

Lines 237-238, Please check whether the data in table 2 is correct.

Lines 304-305, it is not clear why this sentence appears here.

Lines 313-314, I don’t understand this sentence.

Lines 343-345, what are the results of your research? Please check with the relevant results.

Lines 365-368, the comparison between bryophytes and woody plants is of little significance.

Lines 370-372, this sentence contradicts the previous explanation.

Lines 428-560, please supply more latest papers regarding the topics of this study.

Reviewer #2: I. General comments

The authors' manuscript " Ecological stoichiometric characteristics of soil-moss C, N, and P in restoration stages of karst rocky desertification " studied the stoichiometric characteristics of moss and soil. The authors found that soil nutrients in the shrub stage are higher than other restoration stages and the growth of some moss is not affected by the poor rocky desertification soil. Moss could be used as a supplementary method in promoting ecological restoration in areas undergoing karst rocky desertification due to their positive effects on soil nutrients. Those findings may motivate researchers to better understanding the mechanisms of moss in karst areas. It is worthy to publish this manuscript in the journal of " PLOS ONE ". However, this manuscript is not organized very well. It still needs revision.

II. Specific comments

1) Line 48,50,52 (Abstract):" Studying the stoichiometric characteristics of bryophytes and soil", please do not use two different ways to express moss in manuscript.

2) Line 64-66: Please revise the expression. The Abstract should be needs to be condensed.

3) Line 99: Please add the citations.

4) The Introduction should be further introduced some research on the stoichiometry of moss.

5) Line 228-229: Please revise the expression.

6) Line 257:” related to P. C and N had significant negative correlations”, please add appropriate punctuation marks to make sentence correct.

7) Line 287-293: Please revise the expression.

8) Line 293-294,297-298,331-332,329,340-342,364-365 and 392-394: Please add the citations.

9) Line 299-300: Please revise the expression.

10) Line 331-333Pay attention to their logical relationships.

11) Line 335-337These lines look like conclusion? Shouldn't be in the discussion?

12) The Conclusion should be needs to be condensed.

13) Figures

If you can add pictures of moss in karst rocky desertification areas, it will be more helpful to readers understand the study background.

6. PLOS authors have the option to publish the peer review history of their article (what does this mean?). If published, this will include your full peer review and any attached files.

Reviewer #1: No

Reviewer #2: No

---

## [Author Response · Author response to Decision Letter 0]

1 May 2021

[April 11 2021]

Dear Editor:

Thank you for your valuable comments on my article. I have revised them one by one according to expert opinion. The content marked in red in the manuscript is the revised part. The manuscript has been polished by English professional.

The modification is explained as follows:

Reviewer #1:

1. Introduction:

Although the author introduced in detail why ecological restoration should be carried out in karst areas, the current manuscript did not clearly explain why ecological stoichiometric characteristics of bryophytes and soil should be studied in different ecological restoration stages. Why this study is important now and novel compared to the previous research? Similarly, what are the scientific questions of this study? In short, I am not clear about the innovative and scientific purpose of this study.

Answer

We accept expert opinions. 

The introduction has been rewritten to supplement references on moss stoichiometry.

Revision

Introduction

Karst rocky desertification is a process of land degradation caused by the combined effects of natural factors and human activities in the fragile karst background of the subtropics [1-2]. It manifests as the destruction of vegetation, soil erosion, decline in land productivity, and large areas of bare rock, similar to desertified landscapes [3-4]. Rocky desertification [5] has become one of the obstacles to sustainable ecological development in Southwest China [6-7]. In recent years, local conflicts between people and land have been alleviated, and the economy has developed through the planting of economic tree species [8]. However, with tree growth, nutrient requirements have increased annually, leading to diminished soil nutrients, which in turn deteriorate the soil environment in rocky desertification areas.

Ecological stoichiometry reflects the nutritional structure and function of an ecosystem by examining the balance between energy and chemical elements (essential elements such as C, N, and P) in the ecosystem [9-11]. In soil stoichiometry, C: N, C:P and N:P ratios are key indicators that reflect the composition of soil organic matter and the availability of soil nutrients [12-13]. However, the nutrient cycle and ecological stoichiometry of the restoration process in karst rocky desertification are not well understood.

Mosses are one of the most widely distributed plants in the world [14]. Their special leaf surface structure and cell characteristics allow them to withstand high temperatures [15-16] and drought, provide strong water storage capacity and moisture retention ability and stabilize soils [17-18]. Mosses play an important role in preventing and controlling soil erosion on rock surfaces [19]. The H2CO3 formed by moss respiration and secretions can dissolve rocks and form primitive soil [20-22]. Additionally, organic matter secreted by mosses complexes with mineral ions and forms insoluble matter [23-24]. Insoluble matter adheres to moss residue, which not only increases soil deposition but also promotes organic matter accumulation and increases soil nutrient contents [23,25-26]. Compared to bare soil, moss biocrusts were found to have a positive effect on all soil nutrients and to buffer the negative effects of karst rocky desertification, significantly increasing soil microbial richness [27]. Mosses are more sensitive to environmental changes than other plants and are often used for environmental monitoring [28].

Therefore, studying the stoichiometric characteristics of mosses and soil can reveal the nutrient cycle of topsoil during the natural restoration of areas that have undergone karst rocky desertification. This research provides new ideas and methods for controlling karst rocky desertification. Bare rocks, grasslands, shrubs, and secondary forests in different stages of natural restoration following karst rocky desertification were selected as the study area. Mosses and soil were collected from the study sites to detect C, N, and P contents, and the ecological stoichiometric characteristics were analysed.

2.Material and methods:

The authors spent a lot of effort to describe the materials and methods of this study, but unfortunately, the key contents were not shown in the MS. As a reader, I would like to know more about the plant communities in different stages of ecological restoration, such as the geographic information, dominant species, etc. Additionally, it is surprising that the author did not describe the depth of the soil samples collected. These make it difficult for me to make a proper judgment on the results of the study.

Answer

We accept expert opinions. The material and methods have been rewritten, adding missing references.

Revision

Experimental design and field sampling

Bare land, grasslands, shrubs, and secondary forests in restoration areas of karst rocky desertification were chosen as plots（Table.1. Each plot consisted of a circle 400 m2 in size. Mosses were randomly distributed in each plot (Fig.2). According to the moss collection method, each plot was divided into 125 small plots (circles 400 cm2 in size). All the mosses and 1 cm of soil under the mosses in the small plots were collected. The collection number, time, habitat information and characteristics of the mosses were recorded. A total of 500 moss specimens and 500 soil specimens were collected.

Table 1 Habitat characteristics of the plot

different restoration stages latitude and longitude slope aspect rock exposure rate vascular plant species in the plot

bare rock E 105°45′12″，N 26°22′03″ 20° E 90% Celastrus gemmatus、Paederia foetida、Melia azedarach、Fallopia multiflora、Phytolacca americana、Toxicodendron vernicifluum、Setaria viridis、Celtis sinensis、Mahonia fortunei、Solanum americanum、Cyclosorus parasiticus、Asplenium trichomanes、Drynaria roosii

grassland E 105°21′50″，N 26°22′16″ 30° S 40% Toona sinensis、Alangium chinense、Fallopia multiflora、Rubus corchorifolius、Miscanthus sinensis、Justicia procumbens、Ageratum conyzoides、Pteris multifida、Pilea cavaleriei、Talinum paniculatum、Mosla scabra、Mentha canadensis、Digitaria sanguinalis

shrub E 105°45′04″，N 26°22′03″ 60° SE 50% Ilex macrocarpa、Alangium chinense、Ageratina adenophora、Cuscuta chinensis、Debregeasia orientalis、Fissistigma chloroneurum、Akebia trifoliata subsp. australis、Rubus ellipticus、Melia azedarach、Smilax china、Caesalpinia crista、Rosa cymosa、Dalbergia assamica、Litsea coreana var. sinensis、Rubus rosifolius、Rubus tephrodes、Neolepisorus fortunei、Pyrrosia calvata、Pilea cavaleriei、Drynaria roosii、Cladrastis platycarpa、Celtis sinensis、Lindera communis、Nandina domestica、Rhus chinensis、Rhynchosia volubilis

secondary forest E 105°45′04″，N 26°22′03″ 40° SW ，60% Castanea mollissima、Cladrastis platycarpa、Nandina domestica、Broussonetia papyrifera、Rhus chinensis、Celtis sinensis、Caesalpinia crista、Mallotus repandus var. chrysocarpus、Bauhinia championii、Fallopia multiflora、Zanthoxylum dissitum、Pyracantha fortuneana、Debregeasia orientalis、Digitaria sanguinalis、Litsea coreana var. sinensis、Sporobolus fertilis、Triadica sebifera、Rosa cymosa、Akebia trifoliata、Miscanthus sinensis、Ageratina adenophora、Toona sinensis、Lespedeza cuneata、Parthenocissus tricuspidata

Fig.2 The map of the ecosystem of Puding karst rocky desertification ecosystem observation and research station of the chinese academy of sciences. A show the habitat of karst rocky desertification. B shows mosses in the karst rocky desertification habitat.

Determination of C, N and P elements in soil and mosses

The soil samples collected in this study were analysed to determine the SOC, TN, TP and available phosphorus (Olsen-P). SOC was measured using the external heating method. To determine the SOC content, potassium dichromate and sulfuric acid solution were added to air-dried soil samples. The solution was heated in a pan with paraffin oil, boiled for 5 minutes, and titrated with ferrous sulfate solution after cooling, after which the amount of consumed ferrous sulfate was determined. From the amount of consumed ferrous sulfate, the SOC content was calculated. The contents of TN, TP and available phosphorus (Olsen-P) were measured by using previously published methods described by Bao [29].

 The C in the mosses was directly determined by an elemental analyser (Elementar Vario TOC, Germany). Both N and P were digested by concentrated sulfuric acid and hydrogen peroxide. The moss samples were digested in concentrated sulfuric acid until the sulfuric acid decomposed and white smoke was emitted. When the solution was brown and black, it was cooled, hydrogen peroxide was added, and the mixture was boiled for 20 minutes. This process was repeated several times until the digestion liquid was colourless and clear. The liquid was then heated for another 10 minutes to remove excess hydrogen peroxide and cooled to a constant volume. A flow injection instrument (AA3) was used for direct measurement.

3.Results:

There are two serious problems. One is that the language is lengthy, and the description of results still unclear. Another one is interesting that the N and P concentrations were higher C concentrations in the bryophytes under all stages of ecological restoration. Although bryophytes have unique biological characteristics compared with other plants, the current results are not convincing. Of course, if it was not for the miscalculation, I would like the authors will have an appropriate explanation for this phenomenon in the discussion section.

Answer

We accept expert opinions. The results have been rewritten, adding missing references.

Revision

Results

Soil C, N, and P contents and their stoichiometric ratios in different vegetation restoration stages

The average contents of SOC, TN and TP were 133.35 g/kg, 9.99 g/kg, and 0.06 g/kg, respectively. Soil nutrients gradually accumulate with the restoration of karst rocky desertification. However, the contents of SOC, TN, and TP in the shrubland were the highest, and they were significantly different from those in the bare rock, grassland and secondary forest sites (P<0.05). The average content of Olsen-P was 0.62 g/kg, and its change trend was different from that of TP. The highest content of Olsen-P was found in the bare rock habitat (0.732±0.25 g/kg), and this value was significantly different from that in the other three habitats (P<0.05).

Table 1. The contents and stoichiometric ratios of soil C, N, and P in different stages of restoration from karst rocky desertification.

Different lowercase letters in the same column indicate significant differences between different habitats (P<0.05), and P in the stoichiometric ratio represents TP.

The average C:N ratio of soil in the rocky desertification areas was 14.83. With ecosystem succession, the C:N ratio gradually decreased. The C:N ratio in the bare rock site was as high as 19.01±7.61, which was significantly different from that observed for the other three restoration stages (P<0.05). The trends of the changes in C:P and N:P with habitat were basically the same, and the ratios of the bare rock and grassland habitats were significantly different from those of the other habitats (P<0.05).

The contents of C, N, and P in mosses at different vegetation restoration stages and their stoichiometric ratios

The average C, N and P contents of mosses were 3.15 g/kg, 13.97 g/kg and 3.20 g/kg, respectively, in the karst rocky desertification areas (Table 2). The moss C content changed with restoration from karst rocky desertification and was significantly different between different recovery periods (P<0.05). Among the bare rock, grassland, shrub and secondary forest sites, the N and P contents of bryophytes were the highest in the secondary forest, with values of 16.14±3.59 g/kg and 4.04±0.62 g/kg, respectively; these values were significantly different from those of the other sites (P<0.05).

Table 2. The contents and stoichiometric ratios of C, N, and P in mosses at different stages of recovery from karst rocky desertification.

Different lowercase letters in the same column indicate significant differences between different habitats (P<0.05).

The average C:N stoichiometric ratio of the mosses was 0.31. The C:N ratio of the bare rock, shrubs and secondary forests tended to be stable, while that of the grassland was not. The average C:P and N:P stoichiometric ratios of the mosses were 1.42 and 5.91, respectively. Among all the habitats, the C:P and N:P stoichiometric ratios were the highest in the shrubs, and these values and were significantly different from those at other successional stages (P< 0.05). The C:P and N:P stoichiometric ratios in the bare rock, grassland and secondary forest habitats tended to be stable.

The relationships between soil and bryophyte C, N, and P contents and stoichiometric ratios

Correlation analysis revealed that SOC and soil TN were significantly positively correlated with TP and the C:P and N:P ratios. The correlations between soil TP, Olsen-P and the N:P ratio were not significant. TP was negatively correlated with the C:P and C:N ratios. Olsen-P was not correlated with SOC, TN, or the C:P or N:P ratios. The C:N ratio had a significant positive correlation with Olsen-P and significant negative correlations with SOC and TN. There was a significant negative correlation between the C:N and N:P ratios, no correlation between the C:N and C:P ratios, and a significant positive correlation between the N:P and C:P ratios (Table 3).

Table 3. Pearson correlation analysis between C, N, and P contents and stoichiometric ratios of karst rocky desert soil.

There was a significant positive correlation between C and N in mosses, but neither was related to P. C and N had significant negative correlations with the C:N ratio. C had significant positive correlations with the C:P and N:P ratios. P had a significant negative correlation with the C:P and N:P ratios but was not correlated with the C:N ratio. N had a significant positive correlation with the N:P ratio but was not correlated with the C:P ratio. The stoichiometric ratios of the mosses showed a significant positive correlation between the C:P and N:P ratios, a significant negative correlation between the C:N and N:P ratios, and no correlation between the C:P and C:N ratios (Table 4).

The content of C in mosses was significantly positively correlated with SOC (r=0.766, P<0.01), soil TN (r=0.795, P<0.01) and soil TP (r=0.485, P<0.01) but not with soil Olsen-P. The content of N in mosses was significantly positively correlated with SOC (r=0.329, P<0.05) and soil Olsen-P (r=0.338, P<0.05), and there were no significant correlations between the N contents in the mosses and soil N and P contents. The P content in the mosses was significantly positively correlated with soil Olsen-P (r=0.433, P<0.01) but not significantly correlated with soil C, N, or P contents (Fig. 2).

4.Discussion and Conclusion:

The discussion is mostly descriptive or compares the results with others, without much explanation on how the data they use reflects the different ecophysiological process taken place in different stages of ecological restoration.Thus, it is better to give more explanations to the difference of bryophytes (or soil) ecological stoichiometry in different stages of ecological restoration rather than to extensive discussion. So for example, why the soil C, N, and P concentrations were higher in shrub? What are the implications of these results for ecological restoration? Additionally, be concise on your findings and delete unnecessary details and speculative in the Conclusion section.

Answer

We accept expert opinions. The discussion and conclusion have been rewritten, adding missing references.

Revision

Discussion

Soil C, N, and P contents and stoichiometric characteristics of different restoration stages of karst rocky desertification areas

The contents of SOC, soil TN and soil TP in the karst rocky desertification area did not strictly follow the pattern of succession but showed a stepped increasing trend. The contents of SOC, soil TN and soil TP in the shrub habitat were the highest, followed by the contents in the secondary forest. These results are consistent with the research of Li [30]. The consistency between results may be because the pioneer shrubs present during rocky desertification restoration were mainly Rosa cymosa, Rubus corchorifolius, Akebia trifoliata, Cladrastis platycarpa, etc. These plants produce mainly papery leaves, which decompose easily. The pioneer tree species in the secondary forest were mainly Litsea coreana and Castanea mollissima; these species generally produce leathery leaves that take a long time to decompose. The nutrient requirements of arbour tree species are greater than those of other vegetation types [31]. With the restoration of vegetation, the nutrient storage rate of the community decreases, the nutrient cycle accelerates, and the nutrient turnover time is long in the middle-high bud subclimax community stage [32], resulting in the soil nutrient content at the shrub site being significantly higher than that at the secondary forest site.

Notably, this study showed that the average SOC and soil TN contents of mosses in the shrub habitat were 133.35 g/kg and 9.99 g/kg, respectively; these values were significantly higher than the SOC and soil TN in the vegetated areas of the Maolan karst forest (54.72 g/kg and 4.67 g/kg) [33] and karst rocky desertification-affected secondary forests (80.40 g/kg and 2.80 g/kg) [32]. Mosses have a slower decomposition rate than vascular plants, resulting in high organic matter content in moss substrates [34]. Moreover, mosses can form symbiotic relationships with blue algae [35]. Moss-cyanobacteria symbiosis can lead to more efficient N fixation and transport on the soil surface of forests [36], resulting in a high N content in the moss substrate. N fixed by mosses is an important pathway of N sources and sinks for forest ecosystems [37], which has ecological significance that cannot be ignored for ecosystems and even the global nitrogen input and cycle [38].

The soil C:N ratio reflects the soil fertility level and the decomposition rate of soil organic matter [39]. A lower C:N ratio represents high fertility and faster C and N mineralization rates [40]. This study found that the secondary forests and shrubs had lower soil C:N ratios than the other habitat types. The soil C:N ratio reflects the level of soil fertility and the decomposition rate of soil organic matter [41]. Generally, a lower soil C:N ratio represents high fertility and faster C and N mineralization [42]. This shows that under the natural restoration of rocky desertification, soil fertility gradually increases with the succession of the ecosystem. The availability of P is determined by the decomposition rate of soil organic matter, and a lower C:P ratio is an indicator of higher P availability [43]. During the process of restoration from rocky desertification, the C:P ratio tends to increase with succession. The availability of P gradually decreases with succession in the ecosystem. This may be due to the increase in biodiversity as succession advances and the composition of soil nutrients becoming more complicated, both of which limit the availability of P. Therefore, ways to improve soil fertility and promote the sustainable development of soil productivity should be considered in the comprehensive management of rocky desertification via ecological restoration.

C, N, and P contents and stoichiometric characteristics of mosses in karst rocky desertification areas

The differences in the stoichiometric ratios of the key nutrient elements, such as C and N and P, in biomass can regulate and affect the process of carbon consumption or fixation in an ecosystem [44]. The change trend of the C contents in mosses is consistent with the SOC contents in karst rocky desertification areas and significant differences between different recovery stages. However, the P contents in the mosses did not differ greatly between the different recovery stages.

It is worth noting that the average P and N contents in the mosses (3.20 g/kg and 13.97 g/kg, respectively) were higher than the P and N contents of other plants (0.30 g/kg and 6.96 g/kg, respectively) in karst areas [45]. Studies have found that the absorption of P by plants is limited by water, and the use of water is limited by P [46]. Leaf P content is significantly positively correlated with annual average precipitation and precipitation during the growing season [47]. There is no waxy cuticle on the leaves and stems of mosses, and the moss surface can absorb water, nutrients and other substances in the atmosphere [48]. The dry and wet deposition of N and P elements in the atmosphere is also the reason why the N and P contents of mosses are higher than those of other plants. There are differences in the characteristics of the leaf P composition of different plants and functional groups [49]. The morphological structure of mosses is different from that of other plants, and mosses may have special regulatory physiological processes for P.

The growth rate hypothesis posits that changes in the growth rate cause changes in the stoichiometric C, N, and P ratios of organisms [50]. Plants with high growth rates usually have lower C:N, C:P, and N:P ratios, so P is allocated to ribosomal RNA to meet the requirements for rapid synthesis of protein by ribosomes to support rapid plant growth [51-52]. Mosses are different from other plant groups in photosynthetic C fixation and the demand and utilization efficiency of nutrient elements [53]. It is necessary to study the ecological stoichiometric characteristics of moss C, N, P and other elements and establish a theoretical system applicable to moss ecological stoichiometry.

Conclusion

During the natural restoration of karst rocky desertification, SOC and TN contents accumulate with succession. Soil nutrients are higher in areas dominated by shrubs than in other succession stages. The C:N, C:P and N:P stoichiometric ratios increase with the succession of the ecosystem and tend to be stable at the sub-climax community stage. The contents of C, N and P in mosses and their substrates are higher than those in vascular plants and their substrates. The application of mosses could be used as a supplementary method to control karst rocky desertification and promote the sustainable development of the local economy due to their positive effects on improving soil nutrients.

Reviewer #2

The authors' manuscript " Ecological stoichiometric characteristics of soil-moss C, N, and P in restoration stages of karst rocky desertification " studied the stoichiometric characteristics of moss and soil. The authors found that soil nutrients in the shrub stage are higher than other restoration stages and the growth of some moss is not affected by the poor rocky desertification soil. Moss could be used as a supplementary method in promoting ecological restoration in areas undergoing karst rocky desertification due to their positive effects on soil nutrients. Those findings may motivate researchers to better understanding the mechanisms of moss in karst areas. It is worthy to publish this manuscript in the journal of " PLOS ONE ". However, this manuscript is not organized very well. It still needs revision.

Answer

We accept expert opinions. The manuscript has been revised in accordance with experts.

II. Specific comments

1) Line 48,50,52 (Abstract):" Studying the stoichiometric characteristics of bryophytes and soil", please do not use two different ways to express moss in manuscript.

Answer

We accept expert opinions. Moss is used uniformly in the manuscript.

2) Line 64-66: Please revise the expression. The Abstract should be needs to be condensed.

Answer

We accept expert opinions. The abstract has been revised in accordance with experts.

Abstract: 

Rocky desertification is the most serious ecological disaster in karst areas. Comprehensive control of rocky desertification plays an important role in promoting the economic development of karst areas. Studying the stoichiometric characteristics of mosses and soil can provide a powerful reference for the ecological restoration and evaluation of ecosystems experiencing rocky desertification. Soil and mosses were collected from sites representing different stages of ecological restoration (bare rock, grassland, shrubland, and secondary forest), and the contents of carbon (C), nitrogen (N), and phosphorus (P) were detected for ecological stoichiometric analysis. The results indicate that in different restoration stages following karst rocky desertification, the contents of soil organic carbon (SOC), total nitrogen (TN), and total phosphorus (TP) and the stoichiometric ratios in the shrub habitat are higher than those in the bare rock, grassland, and secondary forest habitats. However, the TP and available P contents were low at all stages (0.06 g/kg and 0.62 mg/kg, respectively). The N and P contents and stoichiometric ratios in the mosses showed no significant differences among the succession stages. The C contents in the mosses had a significant positive correlation with SOC and TN and TP content, and the P content had a significant positive correlation with the soil available P. However, there was a significant negative correlation between the C: N and C:P ratios of the bryophytes and soil C: N. In summary, during the process of natural restoration of karst rocky desertification areas, SOC and soil TN contents accumulate with each succession stage. Soil nutrients are higher in shrub habitats than in other succession stages. Mosses have a strong effect on improving soil nutrients in rocky desertification areas.

3) Line 99: Please add the citations.

Answer

We accept expert opinions.

The introduction has been rewritten, adding missing references.

4) The Introduction should be further introduced some research on the stoichiometry of moss.

Answer

We accept expert opinions. 

The introduction has been rewritten to supplement references on moss stoichiometry.

Introduction

Karst rocky desertification is a process of land degradation caused by the combined effects of natural factors and human activities in the fragile karst background of the subtropics [1-2]. It manifests as the destruction of vegetation, soil erosion, decline in land productivity, and large areas of bare rock, similar to desertified landscapes [3-4]. Rocky desertification [5] has become one of the obstacles to sustainable ecological development in Southwest China [6-7]. In recent years, local conflicts between people and land have been alleviated, and the economy has developed through the planting of economic tree species [8]. However, with tree growth, nutrient requirements have increased annually, leading to diminished soil nutrients, which in turn deteriorate the soil environment in rocky desertification areas.

Ecological stoichiometry reflects the nutritional structure and function of an ecosystem by examining the balance between energy and chemical elements (essential elements such as C, N, and P) in the ecosystem [9-11]. In soil stoichiometry, C: N, C:P and N:P ratios are key indicators that reflect the composition of soil organic matter and the availability of soil nutrients [12-13]. However, the nutrient cycle and ecological stoichiometry of the restoration process in karst rocky desertification are not well understood.

Mosses are one of the most widely distributed plants in the world [14]. Their special leaf surface structure and cell characteristics allow them to withstand high temperatures [15-16] and drought, provide strong water storage capacity and moisture retention ability and stabilize soils [17-18]. Mosses play an important role in preventing and controlling soil erosion on rock surfaces [19]. The H2CO3 formed by moss respiration and secretions can dissolve rocks and form primitive soil [20-22]. Additionally, organic matter secreted by mosses complexes with mineral ions and forms insoluble matter [23-24]. Insoluble matter adheres to moss residue, which not only increases soil deposition but also promotes organic matter accumulation and increases soil nutrient contents [23,25-26]. Compared to bare soil, moss biocrusts were found to have a positive effect on all soil nutrients and to buffer the negative effects of karst rocky desertification, significantly increasing soil microbial richness [27]. Mosses are more sensitive to environmental changes than other plants and are often used for environmental monitoring [28].

Therefore, studying the stoichiometric characteristics of mosses and soil can reveal the nutrient cycle of topsoil during the natural restoration of areas that have undergone karst rocky desertification. This research provides new ideas and methods for controlling karst rocky desertification. Bare rocks, grasslands, shrubs, and secondary forests in different stages of natural restoration following karst rocky desertification were selected as the study area. Mosses and soil were collected from the study sites to detect C, N, and P contents, and the ecological stoichiometric characteristics were analysed.

5) Line 228-229: Please revise the expression.

Answer

We accept expert opinions.

The contents of C, N, and P in mosses at different vegetation restoration stages and their stoichiometric ratios

The average C, N and P contents of mosses were 3.15 g/kg, 13.97 g/kg and 3.20 g/kg, respectively, in the karst rocky desertification areas (Table 2). The moss C content changed with restoration from karst rocky desertification and was significantly different between different recovery periods (P<0.05). Among the bare rock, grassland, shrub and secondary forest sites, the N and P contents of bryophytes were the highest in the secondary forest, with values of 16.14±3.59 g/kg and 4.04±0.62 g/kg, respectively; these values were significantly different from those of the other sites (P<0.05).

6) Line 257:” related to P. C and N had significant negative correlations”, please add appropriate punctuation marks to make sentence correct.

Answer

We accept expert opinions. Appropriate punctuation has been added in the manuscript to make sentence correct.

7) Line 287-293: Please revise the expression.

Answer

We accept expert opinions.

Soil C, N, and P contents and stoichiometric characteristics of different restoration stages of karst rocky desertification areas

The contents of SOC, soil TN and soil TP in the karst rocky desertification area did not strictly follow the pattern of succession but showed a stepped increasing trend. The contents of SOC, soil TN and soil TP in the shrub habitat were the highest, followed by the contents in the secondary forest. These results are consistent with the research of Li [30]. The consistency between results may be because the pioneer shrubs present during rocky desertification restoration were mainly Rosa cymosa, Rubus corchorifolius, Akebia trifoliata, Cladrastis platycarpa, etc. These plants produce mainly papery leaves, which decompose easily. The pioneer tree species in the secondary forest were mainly Litsea coreana and Castanea mollissima; these species generally produce leathery leaves that take a long time to decompose. The nutrient requirements of arbour tree species are greater than those of other vegetation types [31]. With the restoration of vegetation, the nutrient storage rate of the community decreases, the nutrient cycle accelerates, and the nutrient turnover time is long in the middle-high bud subclimax community stage [32], resulting in the soil nutrient content at the shrub site being significantly higher than that at the secondary forest site.

8) Line 293-294,297-298,331-332,329,340-342,364-365 and 392-394: Please add the citations.

Answer

We accept expert opinions. The discussion has been rewritten, adding missing references.

9) Line 299-300: Please revise the expression.

We accept expert opinions.

Answer

Notably, this study showed that the average SOC and soil TN contents of mosses in the shrub habitat were 133.35 g/kg and 9.99 g/kg, respectively; these values were significantly higher than the SOC and soil TN in the vegetated areas of the Maolan karst forest (54.72 g/kg and 4.67 g/kg) [33] and karst rocky desertification-affected secondary forests (80.40 g/kg and 2.80 g/kg) [32]. Mosses have a slower decomposition rate than vascular plants, resulting in high organic matter content in moss substrates [34]. Moreover, mosses can form symbiotic relationships with blue algae [35]. Moss-cyanobacteria symbiosis can lead to more efficient N fixation and transport on the soil surface of forests [36], resulting in a high N content in the moss substrate. N fixed by mosses is an important pathway of N sources and sinks for forest ecosystems [37], which has ecological significance that cannot be ignored for ecosystems and even the global nitrogen input and cycle [38].

10) Line 331-333Pay attention to their logical relationships.

Answer

We accept expert opinions. This paragraph has been modified.

11) Line 335-337These lines look like conclusion? Shouldn't be in the discussion?

Answer

We accept expert opinions. The discussion has been rewritten.

The soil C:N ratio reflects the soil fertility level and the decomposition rate of soil organic matter [39]. A lower C:N ratio represents high fertility and faster C and N mineralization rates [40]. This study found that the secondary forests and shrubs had lower soil C:N ratios than the other habitat types. The soil C:N ratio reflects the level of soil fertility and the decomposition rate of soil organic matter [41]. Generally, a lower soil C:N ratio represents high fertility and faster C and N mineralization [42]. This shows that under the natural restoration of rocky desertification, soil fertility gradually increases with the succession of the ecosystem. The availability of P is determined by the decomposition rate of soil organic matter, and a lower C:P ratio is an indicator of higher P availability [43]. During the process of restoration from rocky desertification, the C:P ratio tends to increase with succession. The availability of P gradually decreases with succession in the ecosystem. This may be due to the increase in biodiversity as succession advances and the composition of soil nutrients becoming more complicated, both of which limit the availability of P. Therefore, ways to improve soil fertility and promote the sustainable development of soil productivity should be considered in the comprehensive management of rocky desertification via ecological restoration.

12) The Conclusion should be needs to be condensed.

Answer

We accept expert opinions. The conclusion has been rewritten.

Conclusion

During the natural restoration of karst rocky desertification, SOC and TN contents accumulate with succession. Soil nutrients are higher in areas dominated by shrubs than in other succession stages. The C:N, C:P and N:P stoichiometric ratios increase with the succession of the ecosystem and tend to be stable at the sub-climax community stage. The contents of C, N and P in mosses and their substrates are higher than those in vascular plants and their substrates. The application of mosses could be used as a supplementary method to control karst rocky desertification and promote the sustainable development of the local economy due to their positive effects on improving soil nutrients.

13) Figures

If you can add pictures of moss in karst rocky desertification areas, it will be more helpful to readers understand the study background.

Answer

We accept expert opinions. 

The pictures of moss in karst rocky desertification areas added in manuscript.

Fig. 1 A map of the study area and sampling plots

Fig.2 The map of the ecosystem of Puding Karst Rocky Desertification Ecosystem Observation and Research Station of the Chinese Academy of Sciences. A show the habitat of karst rocky desertification. B shows mosses in the karst rocky desertification habitat.

---

## [Decision Letter · Decision Letter 1]

24 May 2021

Ecological stoichiometric characteristics of soil-moss C, N, and P in restoration stages of karst rocky desertification

PONE-D-21-06846R1

Dear Dr. Dai,

We’re pleased to inform you that your manuscript has been judged scientifically suitable for publication and will be formally accepted for publication once it meets all outstanding technical requirements.

Kind regards,

Fuzhong Wu

Academic Editor

PLOS ONE

Additional Editor Comments (optional):

I suggest the authors removed the Map of China. It is not necessary to cite a map here.

Reviewers' comments:

Reviewer's Responses to Questions

**Comments to the Author**

1. If the authors have adequately addressed your comments raised in a previous round of review and you feel that this manuscript is now acceptable for publication, you may indicate that here to bypass the “Comments to the Author” section, enter your conflict of interest statement in the “Confidential to Editor” section, and submit your "Accept" recommendation.

Reviewer #1: All comments have been addressed

Reviewer #2: All comments have been addressed

Reviewer #3: All comments have been addressed

2. Is the manuscript technically sound, and do the data support the conclusions?

Reviewer #1: Yes

Reviewer #2: Yes

Reviewer #3: Yes

3. Has the statistical analysis been performed appropriately and rigorously? 

Reviewer #1: Yes

Reviewer #2: Yes

Reviewer #3: Yes

4. Have the authors made all data underlying the findings in their manuscript fully available?

Reviewer #1: Yes

Reviewer #2: Yes

Reviewer #3: Yes

5. Is the manuscript presented in an intelligible fashion and written in standard English?

Reviewer #1: Yes

Reviewer #2: Yes

Reviewer #3: Yes

6. Review Comments to the Author

Reviewer #1: I appreciate the progress made in explaining the introduction, materials and methods, results, and discussion section. Also, the authors have also made great efforts to address my concerns. I don’t have any further concerns with the revised manuscript.

Reviewer #2: At the revised version authors have addressed many of the questions mentioned in the first round by both reviewers. I think this manuscript could be published after some minor revision.

Some minor revision suggestion as follows:

1. Table 1 the first row, the uppercase of words for properties description. Such as “slope” change into “Slope” …

2. Table 2 the first row, “different” change into “Different”

3. Table 3 the first row, “different” change into “Different”

Reviewer #3: (No Response)

7. PLOS authors have the option to publish the peer review history of their article (what does this mean?). If published, this will include your full peer review and any attached files.

Reviewer #1: No

Reviewer #2: No

Reviewer #3: No

---

## [Editor Report · Acceptance letter]

18 Jun 2021

PONE-D-21-06846R1 

Ecological stoichiometric characteristics of soil-moss C, N, and P in restoration stages of karst rocky desertification 

Dear Dr. Dai:

I'm pleased to inform you that your manuscript has been deemed suitable for publication in PLOS ONE. Congratulations! Your manuscript is now with our production department. 

Kind regards, 

on behalf of

Professor Fuzhong Wu 

Academic Editor

PLOS ONE